# Effects of the Anti-Tumorigenic Agent AT101 on Human Glioblastoma Cells in the Microenvironmental Glioma Stem Cell Niche

**DOI:** 10.3390/ijms22073606

**Published:** 2021-03-30

**Authors:** Deniz Caylioglu, Rieke Johanna Meyer, Dana Hellmold, Carolin Kubelt, Michael Synowitz, Janka Held-Feindt

**Affiliations:** Department of Neurosurgery, University Medical Center Schleswig-Holstein UKSH, Campus Kiel, 24105 Kiel, Germany; denizcaylioglu@gmail.com (D.C.); rieke@meyer-stoeckte.de (R.J.M.); dana.hellmold@uksh.de (D.H.); carolin.kubelt@uksh.de (C.K.); michael.synowitz@uksh.de (M.S.)

**Keywords:** tumor stem-like cells, microenvironment, heterogeneity, chemoresistance, R-(-)-gossypol, temozolomide, glioblastoma, CXCR7, IL-6R

## Abstract

Glioblastoma (GBM) is a barely treatable disease due to its profound chemoresistance. A distinct inter- and intratumoral heterogeneity reflected by specialized microenvironmental niches and different tumor cell subpopulations allows GBMs to evade therapy regimens. Thus, there is an urgent need to develop alternative treatment strategies. A promising candidate for the treatment of GBMs is AT101, the R(-) enantiomer of gossypol. The present study evaluates the effects of AT101, alone or in combination with temozolomide (TMZ), in a microenvironmental glioma stem cell niche model of two GBM cell lines (U251MG and U87MG). AT101 was found to induce strong cytotoxic effects on U251MG and U87MG stem-like cells in comparison to the respective native cells. Moreover, a higher sensitivity against treatment with AT101 was observed upon incubation of native cells with a stem-like cell-conditioned medium. This higher sensitivity was reflected by a specific inhibitory influence on the p-p42/44 signaling pathway. Further, the expression of CXCR7 and the interleukin-6 receptor was significantly regulated upon these stimulatory conditions. Since tumor stem-like cells are known to mediate the development of tumor recurrences and were observed to strongly respond to the AT101 treatment, this might represent a promising approach to prevent the development of GBM recurrences.

## 1. Introduction

Glioblastomas (GBMs) are highly malignant primary intracranial tumors that are characterized by rapid and infiltrative progression. Despite great improvements in (micro-) surgery and aggressive upfront treatment, patients’ prognosis remains poor due to prompt relapses and resistance to chemo- and radiotherapy [1]. One major reason for this resistance is the pronounced intra- and intertumoral heterogeneity of GBMs [2,3]. Based on various genetically determined morphological properties and protein expression patterns [4], as well as on microenvironmental features like the contribution of different stromal cells [5,6], evolutionary processes within the heterogeneous tumor mass give rise to specialized tumor cell subpopulations (e.g., GBM tumor stem cells, more differentiated GBM cells or fast proliferating or migrating ones) [7,8,9]. The different tumor cell subpopulations communicate either directly or indirectly with e.g., tunneling nanotubes, multiple kinds of extracellular vesicles, or several (inflammatory) mediators like cytokines and chemokines [10,11,12]. They evolve in and adapt to their conquered microcompartment and, as a result, manage to evade, e.g., from suitable treatment schedules [13]. Thus, there is an urgent need to develop alternative treatment strategies in order to overcome the heterogeneity-based chemoresistance of GBMs. Besides the preferred chemotherapeutic drug temozolomide (TMZ), a promising candidate for the treatment of GBMs is AT101, the R(-) enantiomer of the naturally occurring cottonseed-derived polyphenol gossypol [14,15,16,17,18,19,20].

AT101 is characterized by versatile modes of action resulting in, e.g., anti-inflammatory, anti-oxidative, or especially anti-tumorigenic effects (alone or in combination with other drugs) on several tumor entities, including GBMs [16,17,18,19,20,21]. AT101 was able to trigger autophagic, mitophagic, or apoptotic cell death by competitive inhibition of Bcl-2 proteins [22,23,24,25]. A combined gossypol/TMZ treatment resulted in the inhibition of tumor-associated angiogenesis, invasion, and proliferation, as well as in enhanced cytotoxicity on GBMs [16,17]. Sequentially applied single and combined TMZ and AT101 treatment strategies exhibited higher cytotoxicity and better tumor growth control in comparison to single TMZ treatment in a GBM in vitro model [18]. Furthermore, gossypol significantly reduced clonogenic growth of radiation affected GBM cells [22]. The cellular effects of AT101 (alone or in combination with other drugs) were mediated, e.g., by inducing a mitochondrial dysfunction [25], or by interfering with the phosphatidylinositol 3-kinase—Akt and mitogen-activated protein kinase—extracellular-signal-regulated kinase (ERK) signaling pathways [19,26]. Interestingly, AT101 was also able to suppress the growth of TMZ-resistant GBM tumorspheres [27] and could target glioma stem-like cells via the inhibition of the hedgehog and notch signaling pathways resulting in the inhibition of tumor growth [28,29]. However, to date, the influence of the GBM stem cell microenvironment on the AT101 responsiveness of native, non-stem GBM cells is not known. Thus, as a first step, we compared the effects of AT101, alone or in combination with TMZ, on GBM stem-like cells and native, non-stem cells of two different human GBM cell lines by cytotoxicity assays and growth analysis. Secondly, the role of the stem cell microenvironment was analyzed by stimulating native GBM cells with stem-like cell-conditioned medium and AT101, alone or in combination with TMZ. Finally, we investigated the influence of the chemotherapeutic treatment on the mitogen-activated protein kinase-ERK signaling pathway and the gene expression level of selective cytokine and chemokine ligands/receptors as well as of different stem cell markers in the treated native, non-stem GBM cells.

## 2. Results

### 2.1. Stem-Like U251MG and U87MG GBM Cells Show a High AT101-Chemosensitivity

As a first step, we compared the influence of an AT101, TMZ, or a combinational treatment of both on the survival of native and stem-like U251MG and U87MG GBM cells by cytotoxicity assays (Figure 1A–D) and determination of proliferation (Figure 2A–D) for up to six days, respectively. Stem-like and native GBM cell cultures were established and intensively characterized as described before [17,30,31,32].

In relation to controls, both native U251MG and U87MG cells responded only moderately to solely applied TMZ (50 µM) treatment for up to six days (15–20% dead cells) (Figure 1A,B). This effect could be slightly enhanced by the combinational treatment with TMZ (50 µM) and AT101 (5 µM), especially after six days of treatment (25–35% dead cells). However, AT101 showed nearly no cytotoxic effect in both native GBM cell lines when applied individually. Matching results were found concerning the influence of the different chemotherapeutic agents on the proliferation of the native cells (Figure 2A,B). Whereas an application of only AT101 yielded nearly no anti-proliferative effects, TMZ, as well as the combination of TMZ plus AT101 were able to induce clear anti-proliferative effects after six days of treatment in relation to controls in both native cell lines, respectively.

In contrast, stem-like U251MG and U87MG GBM cells were strongly affected by solely applied AT101 (Figure 1C,D) in comparison to the respective controls. After six days of AT101 treatment, up to 70–90% of stem-like U251MG and U87MG cells died. This cytotoxic effect was not clearly increased any further by the combinational treatment with TMZ plus AT101. Furthermore, solely applied TMZ showed only moderate effects in both stem-like GBM cell lines (30–40% dead cells). Accordingly, AT101, and the combination of TMZ plus AT101 strongly inhibited the proliferation in stem-like cells, whereas solely applied TMZ showed only moderate anti-proliferative effects (Figure 2C,D).

### 2.2. Stem-Like Cell-Conditioned Media Enhance the AT101-Chemosensitivity of Native U251MG and U87MG GBM Cells

To analyze whether the high responsiveness of GBM stem-like cells on an AT101 treatment is determined by the environmental stem cell niche as marked by, e.g., stem cell-secreted and probably autocrine and/or paracrine active factors, we started to produce stem-like cell-conditioned media of stem-like U251MG and U87MG GBM cells and performed cytotoxicity and proliferation assays (Figure 3A–F). Therefore, the corresponding native U251MG and U87MG GBM cells were stimulated with stem-like cell-conditioned media containing AT101, TMZ, or the combination of both for up to six days. Equal media, but without any stem-like cell contact, were used as controls for all stimulations.

Concerning the appropriate dimethyl sulfoxide (DMSO) and media controls, the application of the stem-like cell-conditioned media did not significantly influence the cytotoxic effects of solely applied TMZ in native U251MG and U87MG GBM cells, respectively (15–20% dead cells) (Figure 3A,B). On the contrary, stimulation with stem-like cell-conditioned media with AT101 yielded a strong enhancement of the cytotoxic effects of AT101 on the native U251MG and U87MG cells. Here, one should keep in mind that the amounts of dead cells were calculated as described in Equations (1) and (2) (Section 4.3) after three and six days of stimulation in a cumulative manner, respectively. Further, both cell lines responded to the chemotherapeutic treatment with a time delay of two to three days, resulting in relatively low amounts of dead cells at day three, but in strongly increasing amounts of dead cells at day six, respectively. In detail, comparable to the effects of AT101 measured in the stem-like U251MG and U87MG cells (Figure 1C,D), up to 70–90% of native cells treated with stem-like cell-conditioned media containing AT101 died after six days of stimulation (Figure 3A,B). Thus, the stem-like cell-conditioned media seemed to enhance the cytotoxic potential of the AT101 treatment. However, an enhancement of the cytotoxic activity of AT101 was also observed using media without any stem-like cell contact (control media) (Figure 3A,B). Indeed, depending on the native GBM cells investigated, the application of the control media with solely added AT101 resulted in 15–35% dead cells after six days of treatment (for comparison: normal growth media plus AT101: ~5% dead native U251MG and U87MG cells; Figure 1A,B). Interestingly, no higher toxicity resulted when comparing the amounts of dead native cells for the control samples (without any drugs) treated with normal growth medium (Figure 1A,B) or stem-like cell media without any stem-like cell contact (control media) versus those, which were treated with stem-like cell-conditioned media at the different time points (Figure 3A,B).

Summarizing this point, although the control media itself seemed to be able to enhance the cytotoxic potential of AT101 (15–35% dead cells), the stem-like cell-conditioned media produced by the U251MG and U87MG stem-like cells was clearly more efficient (70–90% dead native cells).

The cell death promoting potential of the stem-like cell-conditioned media was also observed upon stimulation with TMZ plus AT101. Indeed, a further enhancement of the cytotoxic efficiency of the combinational treatment was detectable upon application of the stem-like cell conditioned media in the native U251MG and U87MG cells, respectively (80–95% dead cells). This effect was also observed for the control media, but with a considerably lower potential (15–40% dead cells).

Summarized, the different culture conditions per se did not influence the toxicity against glioblastoma cells, but in the presence of drugs, the stem-like cell media (not conditioned) became clearly more toxic to the cells. Nevertheless, the amounts of dead cells distinctly increased further when AT101 and TMZ plus AT101 were added to the GBM cells together with stem-like cell-conditioned media (Figure 3A,B).

The cytotoxic effects were also reflected by the influence of the different chemotherapeutic agents or their combination, applied either with stem-like cell-conditioned or control media, respectively, on the proliferative potential of the native GBM cells (Figure 3C–F). Measured anti-proliferative effects of solely applied TMZ were comparable to those obtained with normal growth media, irrespective of whether stem-like cell conditioned media or the corresponding control media were used. In contrast, both AT101 and the combination of TMZ plus AT101 were able to induce strong anti-proliferative effects on both native GBM cell lines, respectively. Control media were able to promote the anti-proliferative effects of AT101 and TMZ plus AT101, but stem-like cell-conditioned media were clearly more efficient (Figure 3C–F).

### 2.3. ERK Signaling Is Inhibited Upon AT101 and TMZ Plus AT101 Treatment with Stem-Like Cell-Conditioned Media

As a next step, we performed investigations regarding the influence of the different treatment conditions with stem-like cell-conditioned media on the activation of the ERK signaling pathway in native U251MG and U87MG GBM cells. The experimental setup is in line with the previous analysis and was performed with identical treatment conditions for six days. The obtained phospho-ERK (phospho-p42/44) signals were measured using the Image J software, the signals of the phosphorylated kinases were normalized to GAPDH, and the ratios were plotted for each stimulation respectively (Figure 4A,B, Appendix A).

Whereas, in relation to controls, no clear influence on the phosphorylation/activation of p42/44 was detectable upon solely applied TMZ treatment, irrespective of whether administered with control or stem-like cell-conditioned media, a treatment with AT101 as well as with TMZ plus AT101 yielded strong inhibitory effects on the phospho-p42/44 signals. These effects were very prominent in both examined GBM cell lines after six days of treatment. Especially in native U251MG GBM cells, the application of the chemotherapeutic agents combined with the stem-like cell-conditioned media showed very pronounced effects compared to the corresponding control media (Figure 4A). Even so, a distinct inhibition of the phospho-p42/44 signal upon stimulation with stem-like cell-conditioned media could also be observed in native U87MG cells (Figure 4B).

### 2.4. Expression of Stem Cell Markers, CXCR7, and IL-6R Is Regulated Upon AT101 and TMZ Plus AT101 Treatment with Stem-Like Cell-Conditioned Media

As a next step, we considered whether the native U251MG and U87MG cells did undergo dedifferentiation and responded to the stimulation with stem-like cell-conditioned media with a change of the expression of stem cell markers. Thus, we measured the messenger RNA (mRNA) expression of the stem cell markers krüppel-like factor 4 (KLF4), octamer binding transcription factor (OCT4), and Nestin in native U251MG and U87MG GBM cells treated with solely TMZ, AT101, or the combination of TMZ plus AT101 in stem-like cell-conditioned media including the different control samples (Figure 5). In relation to the corresponding controls, an upregulation of the expression of KLF4 was observed after treatment with solely AT101 or TMZ plus AT101 in both U251MG and U87MG native cells (Figure 5A,B). However, this upregulation was detectable irrespective of whether the cells were treated with stem-like cell-conditioned media or the control stem-like cell media without previous stem-like cell contact. The expression of OCT4 was upregulated upon treatment with solely AT101 or TMZ plus AT101 only in U251 native cells, which was again found for both culturing conditions, respectively, whereas in U87MG native cells, no effects of the treatment conditions on OCT4 expression were observed (Figure 5C,D). However, the results were (with some exceptions) not statistically significant when compared to appropriate DMSO controls. Interestingly, Nestin expression was found to be significantly downregulated upon treatment with solely AT101 or AT101 plus TMZ in comparison to the respective controls in U87MG native cells after three days of treatment, whereas in U251MG, no regulation was observed (Figure 5E,F). However, after six days of treatment of U87MG native cells, the expression levels of Nestin were comparable to the respective controls.

Further, to get an insight into stem cell-secreted and probably paracrine active factors, which determined the environmental stem cell niche and are maybe regulated upon a chemotherapeutic treatment, we firstly analyzed the expression pattern of selected cytokines and chemokines together with their corresponding receptors in native versus stem-like GBM cells. As mentioned above, cytokines and chemokines are extremely powerful mediators of the communication of different tumor cell subpopulations within their microenvironmental niche [10,11,12], and, besides this, are also well known to be involved in various tumor progression processes [33,34,35,36,37].

Since we worked with stem-like cell-conditioned media, the investigated factors must be secreted by the stem-like cells themselves, whereas the corresponding receptors must be found on the treated native GBM cells. Thus, firstly, the mRNA expression of different cytokine and chemokine ligands was analyzed in both U251MG and U87MG GBM stem-like cells (Figure 6A,B). The corresponding receptors, or in the case of potential ‘inverse signaling’ properties [38,39], corresponding transmembrane ligands were determined on native U251MG and U87MG cells (Figure 6C,D). Here, the terminus ‘inverse signaling’ describes the fact that the soluble chemokine ligands chemokine (C-X-C motif) ligand 16 (CXCL16) and chemokine (C-X3-C motif) ligand 1 (CX3CL1) were able to induce anti-apoptotic effects after binding to their transmembrane counterparts in ligand-bearing glioma cells [38,39]. Concerning the gene expression analysis, high ΔC_T_ values correspond to a low gene expression and vice versa, and a ΔC_T_ value of 3.33 corresponds to one order of magnitude lower gene expression.

In U251MG stem-like cells, the chemokine ligands CXCL11, CXCL12, CXCL16, and CX3CL1, as well as the cytokines transforming growth factor beta (TGF-β) and interleukin 6 (IL-6) were all expressed in moderate but clearly detectable amounts (ΔC_T_ values 9–13) (Figure 6A). In U87MG stem-like cells, moderate expression of CXCL11 and CX3CL1 was found (ΔC_T_ values 9–13), whereas CXCL12 and CXCL16 were expressed at lower levels (ΔC_T_ values ~16), and TGF-β and IL-6 showed a clearly higher expression (ΔC_T_ values ~5) (Figure 6B).

In both native U251MG and U87MG GBM cells, CXC receptor 4 (CXCR4) (ligand CXCL12), CXCR6 (ligand CXCL16), and CX3C receptor 1 (CX3CR1) (ligand CX3CL1) were not expressed (Figure 6C,D). However, CXCR7 (ligands CXCL11 and CXCL12), CXCL16, and CX3CL1 (in terms of ‘inverse signaling’ transmembrane ligands for CXCL16 and CX3CL1, respectively), as well as TGF-β receptor 1 (TGF-βR1) (ligand TGF-β) and IL-6 receptor (IL-6R) (ligand IL-6), were all expressed in considerable amounts. When comparing both native GBM cell types, CXCL16 and CX3CL1 were clearly lower expressed in U87MG native cells (ΔC_T_ values ~15–25), whereas CXCR7 was highly expressed (ΔC_T_ values ~6) (Figure 6C,D).

Next, to analyze a possible change in the expression of cytokine and chemokine receptors or transmembrane signaling ligands upon chemotherapeutic treatment with the stem-like cell-conditioned media in the native GBM cells, the mRNA of the cells, which were utilized to determinate the cytotoxic and anti-proliferative effects (Figure 3), was isolated and used for further investigations.

Regarding the expression of the chemokine receptor CXCR7 as well as of the transmembrane ligands CXCL16 and CX3CL1 (Figure 7A–F), only CXCR7 expression significantly changed upon treatment. Additionally, an influence on the CXCR7 expression seemed to be more pronounced in the native U251MG cells compared to the U87MG GBM cells (Figure 7A,B). Overall, in relation to corresponding preparations with the control media, the expression of CXCR7 was found to be lower upon stimulation with stem-like cell-conditioned media in both native GBM cell lines. In particular, native U251MG cells responded to an AT101 or a combined TMZ plus AT101 treatment with an upregulation of CXCR7 upon stimulation with the control media, and this effect was clearly smaller upon stimulation with stem-like cell-conditioned media (Figure 7A). In detail, the CXCR7 expression accounted only for the 0.13-fold amount that was detected upon AT101 treatment and for the 0.28-fold amount found upon TMZ plus AT101 application compared to the respective controls. Although in the case of U87MG cells, the regulatory effects of the different treatment strategies on the CXCR7 expression were not very prominent, the combined treatment with TMZ plus AT101 applied with stem-like cell-conditioned media resulted in a statistically significant lower upregulation of CXCR7 compared to the control media (0.089-fold of the amount detected with the control media) (Figure 7B).

As mentioned above, the regulation of the expression level of CXCL16 and CX3CL1 neither in the different cell lines nor upon the various treatment conditions reached statistical significance (Figure 7C–F).

In native U87MG cells, which were treated with stem-like cell-conditioned media plus AT101 or TMZ plus AT101, both CXCL16 and CX3CL1 seemed to be downregulated (Figure 7D,F). However, since the basal expression level of these two chemokines was very low in untreated native U87MG cells (ΔC_T_ values ~15–25), these effects were considered as not relevant.

Regarding the expression of TGF-βR1 and IL-6R (Figure 8A–D), TGF-βR1 expression was not regulated statistically significantly in the different cell types and treatment conditions, respectively (Figure 8A,B). Whereas an IL-6 upregulation was detectable with stem-like cell-conditioned media in U251MG cells only after combined treatment with TMZ plus AT101, in U87MG cells, both solely applied AT101 and combined TMZ plus AT101 treatment yielded an IL-6 upregulation under these treatment conditions (Figure 8C,D). Further, IL-6R expression was induced up to 3.59-fold higher amounts upon AT101 treatment in comparison to control media in U87MG cells. This effect was also observed upon stimulation with TMZ plus AT101 with stem-like cell-conditioned media in U87MG GBM cells, however, without any statistically significant effect compared to the control media.

Summarized, in contrast to native U251MG and U87MG GBM cells, the corresponding stem-like cells were characterized by a high AT101-chemosensitivity, alone or in combination with TMZ, reflected by both an induction of cell death and an inhibition of proliferation. Further, stem-like cell-conditioned media was able to enhance the chemosensitivity of native U251MG and U87MG GBM cells upon AT101 and TMZ plus AT101 treatment. These effects were mirrored by inhibition of the ERK signaling pathway and by a regulation of the expression of different stem cell markers, the chemokine receptor CXCR7 and the cytokine receptor IL-6R upon AT101 and TMZ plus AT101 treatment.

## 3. Discussion

The dismal prognosis of GBMs is largely attributed to the heterogeneous nature of this highly malignant brain tumor. Here, heterogeneity among different tumor cells arises within a single tumor as a result of intrinsic molecular and genetic changes and is also influenced by the microenvironmental niche in which the GBM cells reside [6,9]. Interactions between specialized tumor cell subpopulations and their environment through autocrine or paracrine factors promote invasion and growth of the tumor cells and affect their response to therapy [9,40]. Besides several stromal cells (endothelial cells, immune cells, and other parenchymal cells) and more differentiated GBM cells, GBM stem cells are major key players within the development of tumor recurrence and treatment sensitivity [6,9,40]. Thus, an urgent need exists to develop alternative, more effective treatment strategies. In addition to the preferred chemotherapeutic drug TMZ, a promising candidate for the treatment of GBMs is AT101 [14,15,16,17,18,19,20]. AT101 is able to trigger autophagic, mitophagic, or apoptotic cell death by versatile mechanisms of action as the inhibition of Bcl-2 proteins, the induction of mitochondrial dysfunction, or by modulating serval signaling pathways in a specific manner [22,23,24,25,26,41]. Moreover, since it is known that AT101 is able to suppress the growth of glioma stem cells [28,29], AT101 seems to be a promising chemotherapeutic agent to prevent GBM recurrences, which often arise from dysfunctional (tumor) stem cells [6,9,40]. However, to date, the influence of the GBM stem cell microenvironment on the AT101 responsiveness of native, non-stem GBM cells is not known.

In this study, we were able to show that stem-like cells generated from two different GBM cell lines (U251MG and U87MG) responded to treatment with AT101 or TMZ plus AT101 with a pronounced induction of cell death and inhibition of proliferation. These distinct effects were not observed in corresponding non-stem, native U251MG, and U87MG cells but could be induced in native cells upon AT101 or TMZ plus AT101 treatment with stem-like cell-conditioned media. Further, the stem cell markers KLF4 and OCT4 were (slightly) upregulated upon these treatment conditions in the native U251MG and U87MG cells, and this effect was reflected by the inhibition of the ERK signaling pathway.

ERK signaling pathways are well known to be involved in processes of cell survival and proliferation [42]. Thus, a more pronounced inhibition indeed could result in a more distinct induction of cell death and inhibition of proliferation upon AT101 and TMZ plus AT101 treatment in our stem-like cell niche model. For example, Sadahira et al. stated that gossypol was able to trigger cell death by the inhibition of the phosphorylation of JAK2, STAT3, ERK1/2, and p38 kinases in multiple myeloma [43], and Mehner et al. showed that AT101 in combination with desmethoxycurcumin yielded anti-proliferative effects via the reduction of the phosphorylation of the ERK signal in human primary GBM cells [19]. Further, AT101 could target glioma stem-like cells via the inhibition of the hedgehog and notch signaling pathways resulting in the inhibition of tumor growth [28,29]. Thus, enhanced chemosensitivity of AT101 could be triggered by an influence on several different signaling pathways, depending on the specific microenvironmental conditions.

However, an increased AT101-chemosensitivity in the stem cell niche could also be mediated via other mechanisms. For example, gossypol was shown to regulate the membrane microviscosity in adrenocortical carcinoma cells and to induce specific changes on the membrane lipid matrix in adrenal cortex mitochondria [44,45]. In adrenocortical carcinoma cells, the membranes exposed to gossypol became more rigid, and gossypol also increased the microviscosity of isolated mitochondrial and microsomal enriched membrane preparations [44,45]. These properties of gossypol resulted in reduced tumor prevalence in an adrenocortical carcinoma in vivo model. Additionally, gossypol was able to induce a conductance in phospholipid bilayer membranes, which was accompanied by an increase in proton permeability. This effect could be responsible for the inhibitory effects of gossypol on several membrane transport systems [46]. Since diverse cell types differ regarding the composition of their lipid membranes [47], and the microenvironmental niche is naturally involved in such differences, the enhanced AT101-sensitivity in the stem-like cell niche could also be determined by such processes.

Moreover, the interplay of stem-like cell secreted factors and the corresponding receptors on the recipient cells is another important point. Indeed, an influence of stem cell secreted active paracrine factors on the behavior of non-stem cells was previously described. Chen et al. [48] showed that activation of the insulin-like growth factor receptor signaling could induce Nanog expression and promoted stemness in cancer cells in the presence of cancer-associated fibroblasts, and Heddleston et al. [49] documented that a hypoxic microenvironment promoted a reprogramming towards a cancer stem cell phenotype.

Thus, exemplified for a select panel of different chemokines and cytokines, we proved whether the expression of these factors differed upon AT101 or TMZ plus AT101 treatment in native GBM cells when stimulated with stem-like cell-conditioned media. Interestingly, compared to the respective control media samples, a lower upregulation of the CXCL11/12 receptor CXCR7 was determined in the treated native cells. CXCR7 is well known to be upregulated upon chemotherapeutic treatment and, further, to mediate several pro-tumorigenic processes of CXCL11/12 in GBMs like anti-apoptotic effects and promotion of proliferation by, e.g., induction of the ERK signaling pathway [17,31,50]. In accordance with this, Liu et al. [51] presented that the targeting of CXCR7 inhibited glioma cell proliferation and mobility. Thus, a low expression of CXCR7 seems to be in line with the enhanced AT101-chemosensitivity in our stem-like cell niche model.

Contrary, the IL-6R was significantly upregulated in native U251MG and U87MG GBM cells when stimulated with AT101 or TMZ plus AT101 with stem-like cell-conditioned media. IL-6 itself is known to be involved in several processes that favor tumor progression like, e.g., regulation of apoptosis, survival, proliferation, angiogenesis, metabolisms, and invasion of tumor cells [52]. However, an anti-tumorigenic, beneficial role of IL-6 was also previously described. IL-6 had broad effects on leukocyte survival, proliferation, differentiation, and recruitment [53,54], and was able to promote anti-tumor immunity to attain tumor control [55]. Thus, an upregulation of the IL-6R could be either explained by the activation of anti-tumorigenic, more immunological-based processes or as an anti-apoptotic regulatory escape mechanism in order to overcome the induction of cell death by the AT101 or TMZ plus AT101 treatment. Interestingly, when we stimulated native U251MG and U87MG cells in the regular medium with the addition of recombinant human IL-6, the additional application of solely IL-6 did not enhance the AT101 cytotoxicity on native U251MG and U87MG GBM cells upon these treatment conditions. Results are shown in Appendix A. Thus, the role of the IL-6-IL-6R axis upon chemotherapeutic treatment with AT101 in the microenvironmental stem cell niche seems to be complex.

## 4. Materials and Methods

### 4.1. Cultivation of Native and Stem-Like GBM Cells

The human glioblastoma cell lines U87MG (ECACC 89081402) and U251MG (ECACC 89081403; formerly known as U373MG) were obtained from the European Collection of Cell Cultures (ECACC, Salisbury, UK) and cultured under non-stem cell conditions in Dulbecco’s modified Eagle’s medium (DMEM; Thermo Fisher Scientific, Waltham, MA, USA) supplemented with 10% fetal bovine serum (FBS; Thermo Fisher Scientific), 1% penicillin streptomycin (10,000 U/mL; Thermo Fisher Scientific), and 2 mM additional L-glutamine (Thermo Fisher Scientific) as described before [17,30]. Stem-like cell cultures were established and intensively characterized as described before [17,30,31,32] by cultivating U251MG and U87MG cells in neurosphere medium (50% DMEM, 50% F12 medium (Thermo Fisher Scientific) containing the following supplements: 2 mM L-glutamine, 0.6% glucose (Roth, Karlsruhe, Germany), 9.5 ng/mL putrescine dihydrochloride (Sigma-Aldrich, St. Louis, MO, USA), 6.3 ng/mL progesterone (Sigma-Aldrich), 5.2 ng/mL sodium selenite (Sigma-Aldrich), 0.025 ng/mL insulin (Sigma-Aldrich), 2 μg/mL heparin (Sigma-Aldrich), and 4 mg/mL bovine serum albumin (Thermo Fisher Scientific)). The growth factors EGF (epidermal growth factor; PeproTech, Rocky Hill, NJ, USA) and bFGF (basic fibroblast growth factor; ImmunoTools, Friesoythe, Germany) were added at a concentration of 20 ng/mL. Glioma stem-like cells were characterized by the formation of neurospheres, the ability to survive and proliferate under stem cell conditions, and to differentiate into more mature cells, which was proven as described before [30,31,32]. The purity of the GBM cells was ascertained by immunostaining with cell type specific markers, and by the absence of contamination with mycoplasms. GBM cell line’s identity was proven routinely by Short Tandem Repeat profiling at the Department of Forensic Medicine (Kiel, Germany) using the Powerplex HS Genotyping Kit (Promega, Madison, WC, USA) and the 3500 Genetic Analyser (Thermo Fisher Scientific).

### 4.2. Preparation of Stem-Like Cell-Conditioned Media

For the preparation of stem-like cell-conditioned media, 5.0 × 10^4^/mL U251MG and U87MG stem-like cells were cultured over three days in 50% DMEM plus 50% F12 medium with all supplements indicated above. After this, media were changed, and stem-like GBM cells were further cultured over three days in 50% DMEM plus 50% F12 media without phenol-red (Thermo Fisher Scientific) with all supplements mentioned above. Upon completion of the second three days cultivation period, stem-like cell-conditioned media were aspirated and centrifuged (1100 rpm, 6 min, RT); the supernatants were supplemented with 10% FBS (Thermo Fisher Scientific) and used for the stimulation of the corresponding native U251MG and U87MG cells. Control media without any stem-like cell contact were prepared in the same way, including an incubation period over three days at 37 °C in a humidified atmosphere with 5% CO_2_ and addition of FBS directly before the stimulation of the native GBM cells.

### 4.3. Cytotoxicity Assay and Determination of Proliferation

For stimulation of native GBM cells, 1.0 × 10^5^/mL native U251MG and U87MG were seeded in 6-well culture dishes in DMEM (Thermo Fisher Scientific) supplemented with 10% FBS (Thermo Fisher Scientific), 1% penicillin streptomycin (10,000 U/mL; Thermo Fisher Scientific), and 2 mM additional L-glutamine (Thermo Fisher Scientific), and were stimulated after one to two days of cultivation with either 5 μM AT101 [stock dissolved at 100 mM in dimethyl sulfoxide (DMSO); Tocris, Bristol, UK], 50 μM TMZ (stock dissolved at 100 mM in DMSO; Sigma-Aldrich), as well as a combination of 5 µM AT101 and 50 µM TMZ for up to six days in (i) DMEM without phenol-red (PANBiotech GmbH, Aidenbach, Germany) supplemented with 10% FBS (PANBiotech GmbH), 1% penicillin streptomycin and 2 mM additional L-glutamine, with or without human recombinant interleukin-6 (10 ng/mL; PeproTech); (ii) stem-like cell conditioned media supplemented with 10% FBS (Thermo Fisher Scientific); (iii) control media without any stem-like cell contact supplemented with 10% FBS (Thermo Fisher Scientific). Controls were stimulated with the equal volume of DMSO (0.01% (*v*/*v*)). For stimulation of stem-like GBM cells, 1.0 × 10^5^/mL stem-like U251MG and U87MG cells were seeded in 6-well culture dishes in 50% DMEM plus 50% F12 medium (Thermo Fisher Scientific) with all supplements indicated above. Stem-like cells were stimulated after three to four days of cultivation with 5 μM AT101, 50 μM TMZ, or a combination of 5 µM AT101 and 50 µM TMZ for up to six days in 50% DMEM plus 50% F12 media without phenol-red (Thermo Fisher Scientific) with all supplements mentioned above. Controls were stimulated with equal volume of DMSO [0.01% (*v*/*v*)]. Cytotoxic effects of AT101, TMZ, and DMSO treatment on different cell types were investigated with the CytoTox- FluorTM Cytotoxicity Assay (Promega) according to manufacturer’s instruction and as described before [18,19]. Briefly, supernatants of different stimulated and control cells were collected at days three and six of stimulation, mixed with the bis-AAF-R110 substrate, and measured in a fluorescence microplate reader (GENios, TECAN, Zürich, Switzerland) at 485/535 nm. The exact numbers of dead cells were determined according to a prepared standard of digitonin-lysed (82.5 μg/mL; Merck Millipore, Burlington, MA, USA) cell-dilutions of each cell culture, respectively. Furthermore, percentages [%] of dead cells in relation to the total amount of measured cells were calculated as described in Equations (1) and (2) after three and six days of stimulation, respectively.
[%] dead cells [day 3] = (number of dead cells [day 3]/(number of dead cells [day 3] + vital cells [day 3])) × 100(1)
[%] dead cells [day 6] = (number of dead cells [day 3 + day 6]/(number of dead cells [day 3 + day 6] + vital cells [day 6])) × 100(2)

The cell survival was determined by counting viable cells at day zero, three, and six of stimulation. Growth rates were calculated as n-fold number of alive cells compared to day zero of the stimulation, respectively.

### 4.4. Western Blot

For the Western blot, 2.5–3.0 × 10^5^ native U251MG and U87MG were seeded in T25 flasks in DMEM (Thermo Fisher Scientific) supplemented with 10% FBS (Thermo Fisher Scientific), 1% penicillin streptomycin (10,000 U/mL; Thermo Fisher Scientific), and 2 mM additional L-glutamine (Thermo Fisher Scientific), and were stimulated after one day of cultivation with 5 μM AT101, 50 μM TMZ, or a combination of 5 µM AT101 and 50 µM TMZ for up to six days in (i) stem-like cell-conditioned media supplemented with 10% FBS (Thermo Fisher Scientific), or (ii) control media without any stem-like cell contact supplemented with 10% FBS (Thermo Fisher Scientific). Controls were stimulated with equal volume of DMSO [0.01% (*v*/*v*)]. Cells were harvested, and 3 to 10 µg per sample of protein were used for Western blotting experiments as described before [17,19,31,32]. The primary antibody was anti-phospho-p42/44 (1:1,000, #9101, rabbit; Cell Signaling, Danvers, MA, USA) in 5% [*w*/*v*] bovine serum albumin/tris-buffered saline with 0.1% tween (TBS-T), the secondary antibody was donkey anti-rabbit IgG-HRP (1:12,500, A16035; Thermo Fisher Scientific) in 2% [*w*/*v*] casein/TBS-T. Loading of equal amounts of protein was confirmed by stripping and incubation of the membranes with anti-glycerinaldehyde 3-phosphate dehydrogenase (GAPDH) (1:200; sc-47724; mouse, Santa Cruz Biotechnology, Dallas, TX, USA) in 2% [*w*/*v*] casein/TBS-T, the secondary antibody was donkey anti-mouse IgG-HRP (1:10,000, sc-2096; Santa Cruz Biotechnology) in 2% [*w*/*v*] casein/TBS-T as described before [31]. The densities of kinase signals were measured using the Image J software, the signals of phosphorylated kinases were normalized to GAPDH, and the ratios were plotted for each stimulation, respectively.

### 4.5. Reverse Transcription and Quantitative Real-Time PCR (qRT-PCR)

RNA of cells was isolated with the TRIzol^®^ Reagent (Thermo Fisher Scientific) or with the ARCTURUS^®^ PicoPure^®^ RNA Isolation Kit (Applied Biosystems, Waltham, MA, USA) according to the manufacturer’s instructions. DNase digestion, cDNA synthesis, and qRT-PCR were performed as described before [17,18,19,20] using TaqMan primer probes (Applied Biosystems): CXCL11 (Hs00171138_m1), CXCL12 (Hs00171022_m1), CXCL16 (Hs00222859_m1), CX3CL1 (Hs00171086_m1), CXCR4 (Hs00237052_m1), CXCR6 (Hs00174843_m1), CXCR7 (Hs00664172_s1), CX3CR1 (Hs00365842_m1), GAPDH (Hs99999905_m1), interleukin-6 (IL-6) (Hs00985639_m1), interleukin-6 receptor (IL-6R) (Hs01075664_m1), krüppel-like factor 4 (KLF4) (Hs00358836_m1), Nestin (Hs00707120_s1), octamer binding transcription factor (OCT4) (Hs00999632_g1), transforming growth factor-β (TGF-β) (Hs00171257_m1), and TGF-β receptor 1 (TGF-βR1). Cycles of threshold (C_T_) were determined, and ΔC_T_ values of each sample were calculated as C_T gene of interest_ − C_T GAPDH_. The induction of gene expression upon stimulation is displayed as relative gene expression, calculated as n-fold expression changes = 2 ΔC_T control_ − ΔC_T stimulus_.

### 4.6. Statistical Analysis

The data were statistically analyzed using the GraphPad Prism 5 Software (GraphPad Software, San Diego, CA, USA). A two-way analysis of variances (ANOVA) followed by a Tukey’s multiple comparison test was performed, as separately indicated for each experiment. In general, the data are presented as mean ± standard deviation (SD). Statistical significance is marked with asterisks depending on the *p*-value: * *p* < 0.05, ** *p* < 0.01 and *** *p* < 0.001.

## 5. Conclusions

In summary, we showed a high AT101-chemosensitivity of U251MG and U87MG GBM stem-like cells and native cells upon treatment with AT101 with a stem-like cell-conditioned medium. A specific inhibitory influence on the p-p42/44 signaling pathway, as well as a distinct regulation of the expression of individual stem cell markers, of the CXCR7 and the IL-6R upon AT101 and TMZ plus AT101 treatment, was found. Of course, in vivo studies should be performed to further verify the efficiency of a combined TMZ plus AT101 treatment, especially in the stem cell niche. Here, in our opinion, a local application of AT101 directly in the tumor resection cavity (e.g., by nanoparticles or (biodegradable) drug carrier devices) together with systemic treatment with TMZ will be the most promising approach. Using this local treatment strategy of AT101, it is possible to overcome the known relatively low bioavailability of AT101 after systemic treatment [20], and to analyze the potential of AT101 to eradicate TMZ-resistant glioblastoma cells, which in most cases are characterized by pronounced stem-like cell characteristics [17]. Further, since it was shown that AT101 was a competent enhancer of radiation-induced apoptosis in head and neck squamous cell carcinoma in vitro [56], a combination of AT101 and radiation might also be feasible to treat glioblastomas more efficiently. The presented results might be helpful in order to understand the complex influence of microenvironmental tumor cell niches upon chemotherapeutic treatment strategies more precisely.

## Figures and Tables

**Figure 1 ijms-22-03606-f001:**
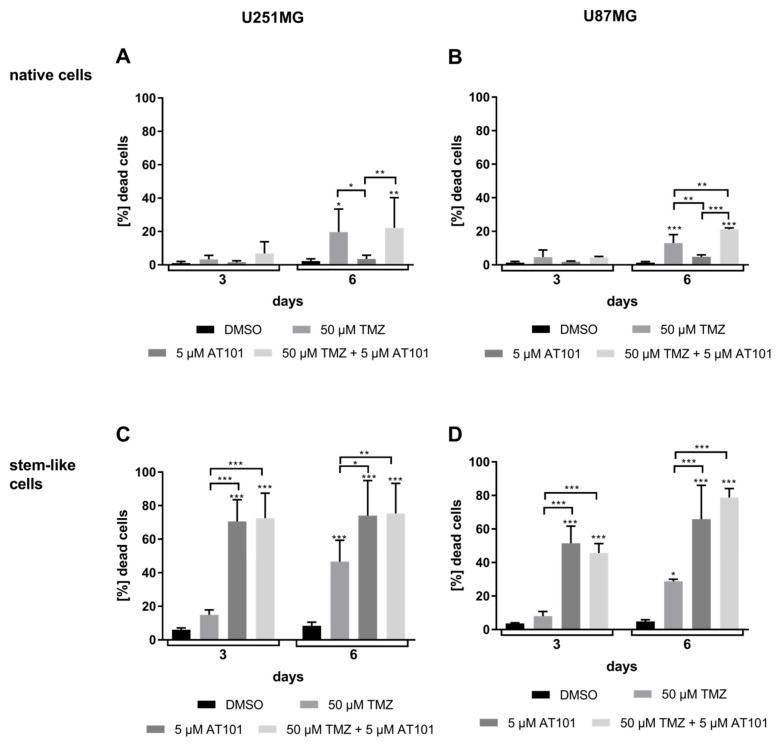
AT101 affected the survival of stem-like U251MG and U87MG glioblastoma (GBM) cells to a greater extent compared to native GBM cells. After stimulation with dimethyl sulfoxide (DMSO) (control), temozolomide (TMZ) [50 μM], AT101 [5 μM], or TMZ [50 μM] and AT101 [5 μM], a cytotoxicity assay was carried out after three and six days, respectively. Death rates of native (**A**) U251MG and (**B**) U87MG cells, as well as of stem-like (**C**) U251MG and (**D**) U87MG cells are depicted. The significances between different stimulation approaches were determined with a two-way analysis of variances (ANOVA) test followed by a Tukey’s multiple comparison test. Significant differences compared to the DMSO control are indicated directly above the bars and between different stimulations on a line linking the bars (* *p* < 0.05; ** *p* < 0.01; *** *p* < 0.001). Error bars correspond to the standard deviation, *n* = 3.

**Figure 2 ijms-22-03606-f002:**
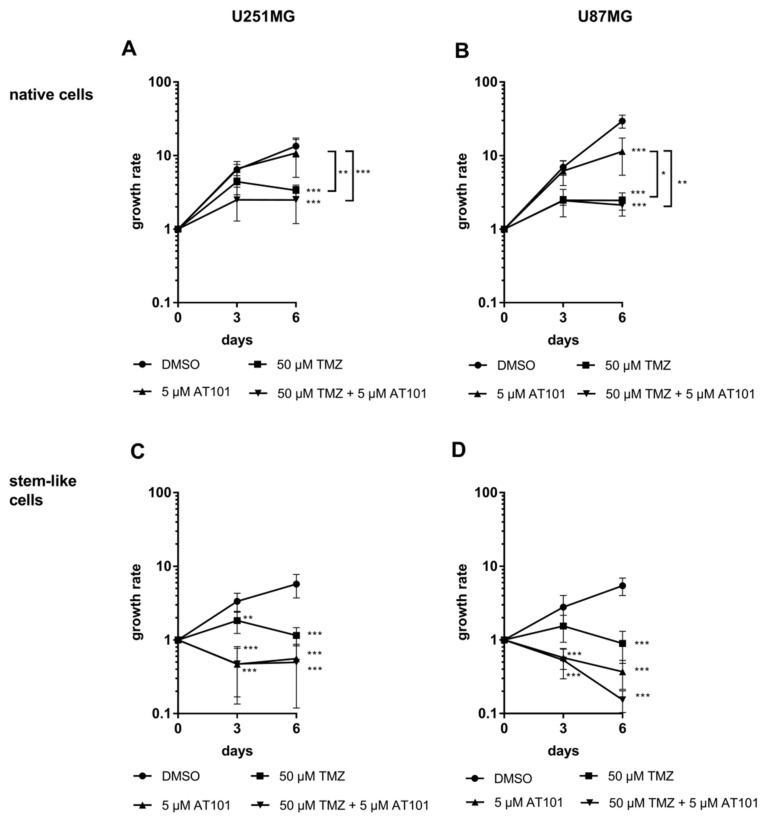
The anti-proliferative effects of AT101 were enhanced in stem-like U251MG and U87MG GBM cells compared to native GBM cells. Cells were stimulated with DMSO (control), temozolomide [50 μM], AT101 [5 μM], or temozolomide [50 μM] and AT101 [5 μM]. Counting of viable cells was performed at day 0, 3, and 6 of stimulation, and growth rates were calculated as the n-fold number of alive cells compared to day 0. Growth curves of native (**A**) U251MG and (**B**) U87MG cells, as well as of stem-like (**C**) U251MG and (**D**) U87MG cells are depicted. The significances between different stimulation approaches were determined with a two-way ANOVA test followed by Tukey’s multiple comparison test. Significant differences compared to the DMSO control are indicated directly next to the curves and between different stimulations on a line linking the curves (* *p* < 0.05; ** *p* < 0.01; *** *p* < 0.001). Error bars correspond to the standard deviation, *n* = 3.

**Figure 3 ijms-22-03606-f003:**
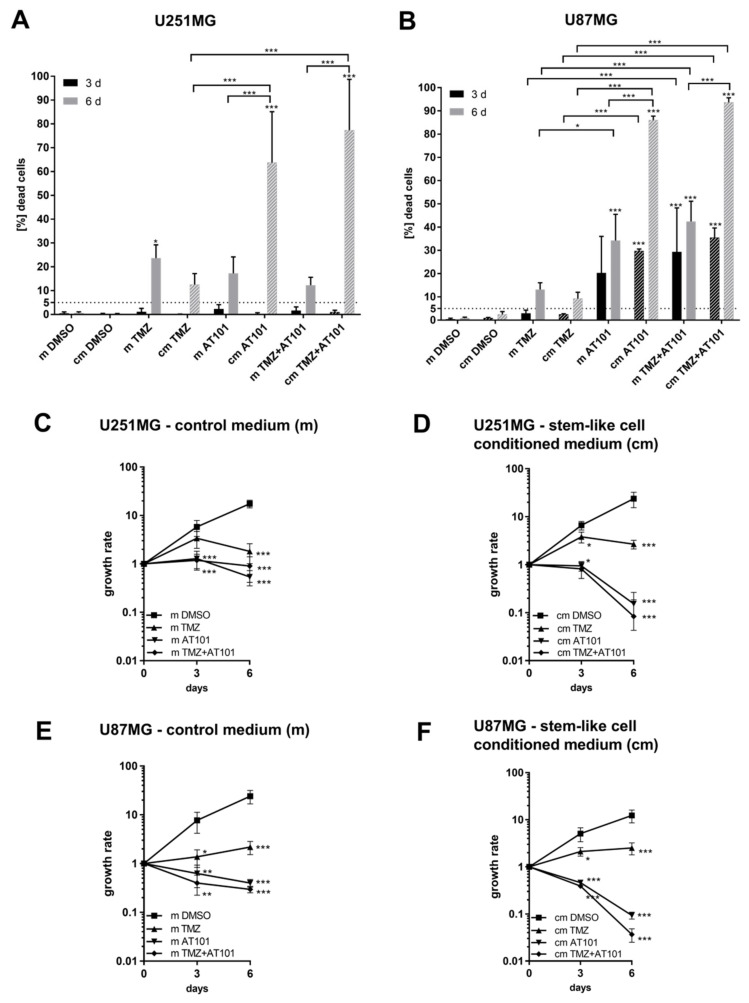
Stimulation with stem-like cell-conditioned media increased the effect of AT101 on the survival and proliferation of native U251MG and U87MG cells. The different cell lines were stimulated with control medium (m) and stem-like cell-conditioned media (cm) containing DMSO, temozolomide [50 μM], AT101 [5 μM], or a combination of temozolomide [50 μM] and AT101 [5 μM]. A cytotoxicity assay was carried out after three and six days, respectively, revealing death rates of native (**A**) U251MG and (**B**) U87MG cells. Cell counts were determined on days 0, 3, and 6 of stimulation. Growth rates were calculated as the n-fold number of alive cells on days 3 and 6 compared to day 0 of stimulation for both U251MG and U87MG cells incubated with (**C**,**E**) control medium or (**D**,**F**) stem-like cell-conditioned medium. The significances between different stimulation approaches were determined with a two-way ANOVA test followed by Tukey’s multiple comparison test. Significant differences compared to the DMSO control are indicated directly above the bars/directly next to the curves and between different stimulations on a line linking the bars/the curves (* *p* < 0.05; ** *p* < 0.01; *** *p* < 0.001). Error bars correspond to the standard deviation, *n* = 3.

**Figure 4 ijms-22-03606-f004:**
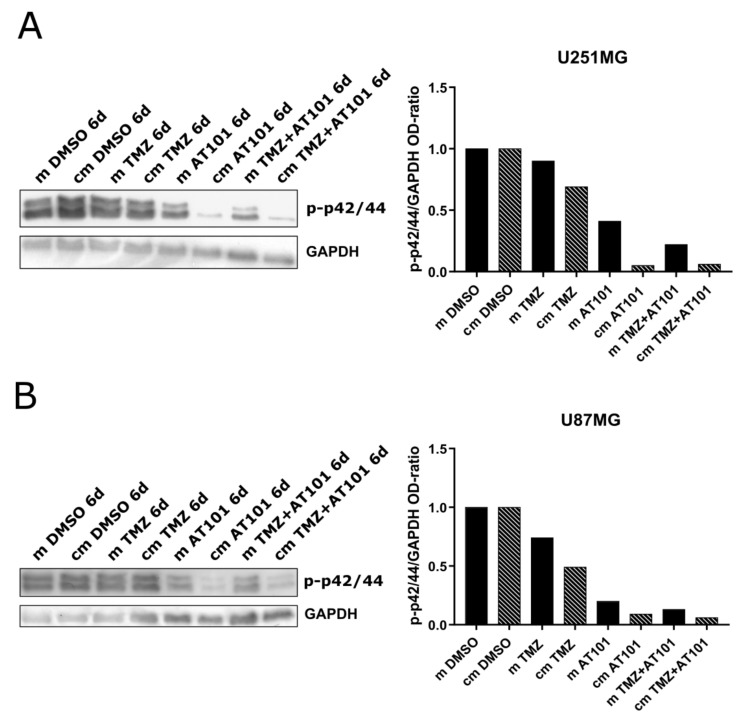
Inhibition of the extracellular-signal-regulated kinase (ERK) signaling upon AT101 and temozolomide (TMZ) plus AT101 treatment with stem-like cell-conditioned media. Western blot on phospho-p44/42 of native (**A**) U251MG and (**B**) U87MG cells after stimulation with control medium (m) or stem-like cell-conditioned medium (cm) containing DMSO, TMZ [50 µM], AT101 [5 µM], or TMZ [50 µM] and AT101 [5 µM] for six days. The obtained phospho-ERK (phospho-p42/44) signals were normalized to glycerinaldehyde 3-phosphate dehydrogenase (GAPDH). Exemplary data shown; *n* = 2.

**Figure 5 ijms-22-03606-f005:**
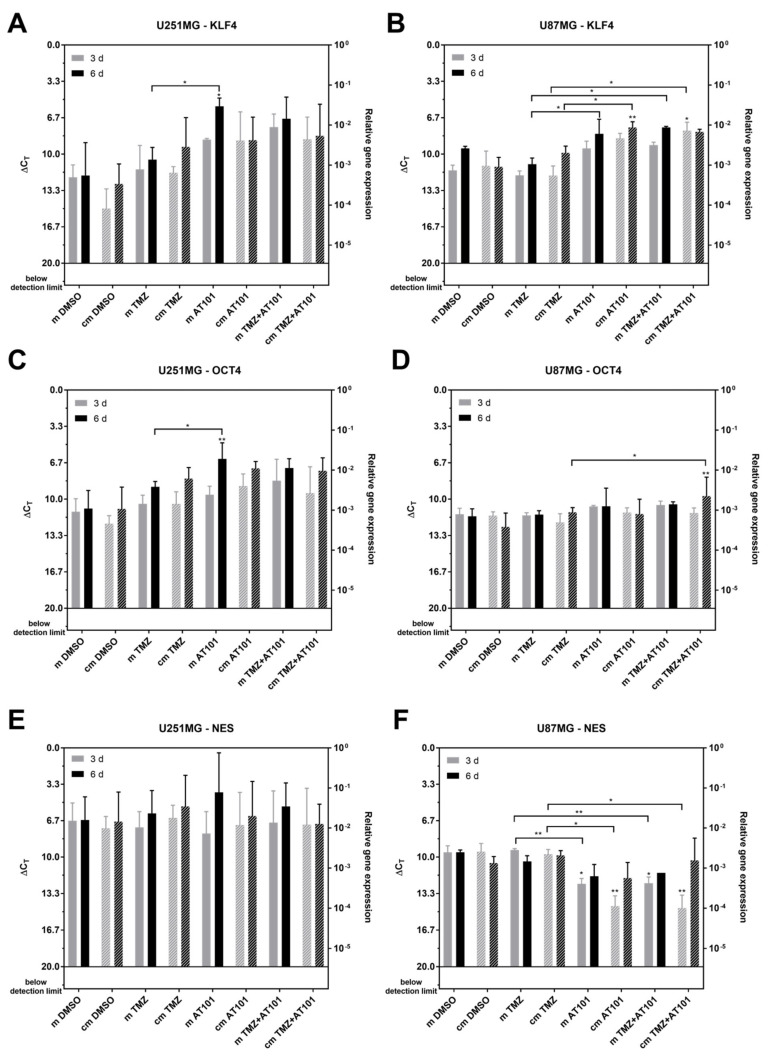
Quantitative PCR (qPCR) analysis of the messenger RNA (mRNA) expression of the different stem cell markers krüppel-like factor 4 (KLF4), octamer binding transcription factor 4 (OCT4), and Nestin (Nes) in native U251MG and U87MG cells after stimulation with control medium (m) or stem-like cell-conditioned medium (cm) supplemented with DMSO (control), TMZ [50 µM], AT101 [5 µM], or TMZ [50 µM] and AT101 [5 µM], respectively, for three and six days. KLF4 expression in (**A**) U251MG cells and (**B**) U87MG cells; OCT4 expression in (**C**) U251MG cells and (**D**) U87MG cells; Nestin expression in (**E**) U251MG cells and (**F**) U87MG cells (*n* = 3). The significances between different stimulation approaches were determined with a two-way ANOVA test followed by Tukey’s multiple comparison test. Significant differences compared to the DMSO control are indicated directly above the bars and between different stimulations on a line linking the bars (* *p* < 0.05; ** *p* < 0.01). Error bars correspond to the standard deviation.

**Figure 6 ijms-22-03606-f006:**
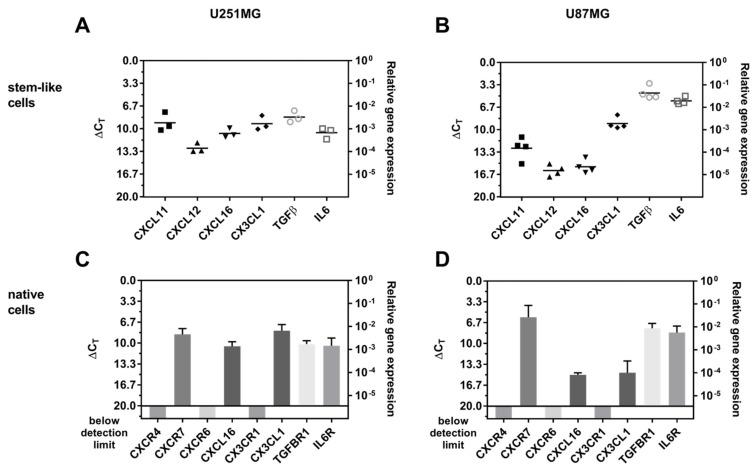
qPCR analysis of the mRNA expression of cytokines and chemokines together with their corresponding receptors in stem-like versus native GBM cells. Gene expression of different cytokine and chemokine ligands in stem-like (**A**) U251MG (*n* = 3) and (**B**) U87MG cells (*n* = 4); Corresponding receptor expression of native (**C**) U251MG cells (*n* = 3) and (**D**) U87MG cells (*n* = 3). The error bars in (**C**,**D**) correspond to the standard deviation.

**Figure 7 ijms-22-03606-f007:**
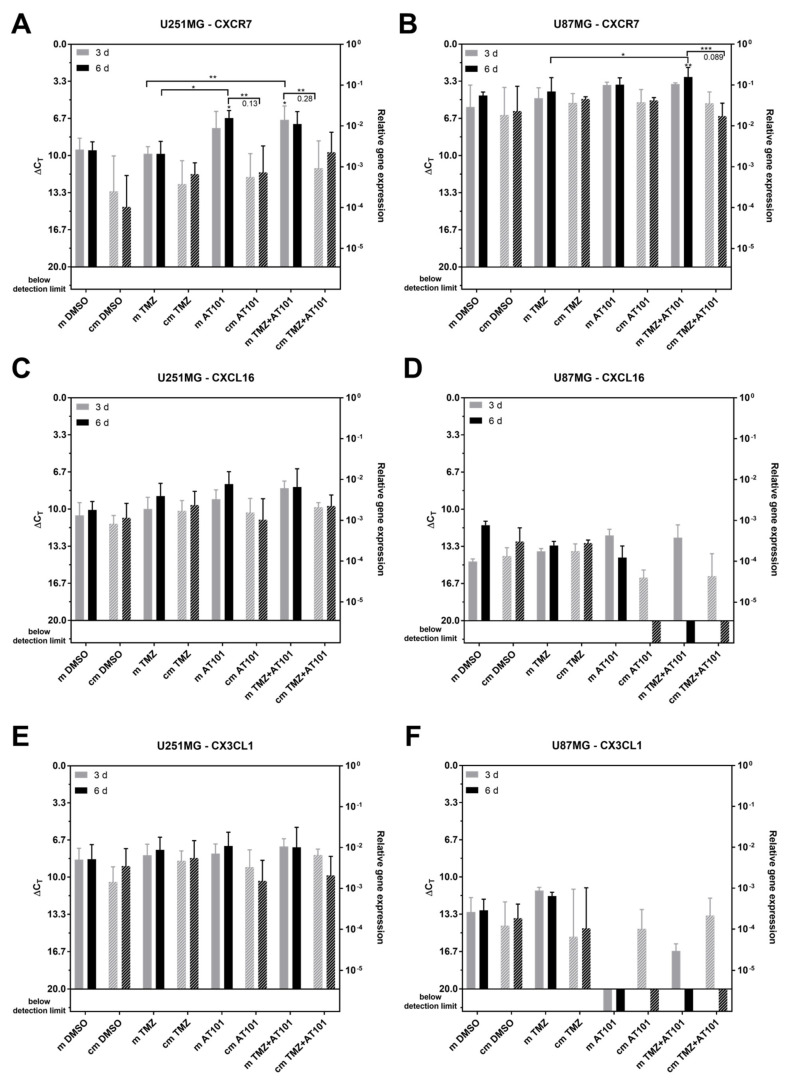
Chemokine (C-X-C motif) receptor 7 (CXCR7) was significantly less upregulated upon AT101 and TMZ plus AT101 treatment with stem-like cell conditioned media in comparison to the control medium. qPCR analysis of the mRNA expression of the chemokine receptor CXCR7 and the transmembrane ligands CXC ligand 16 (CXCL16) and chemokine (C-X3-C motif) ligand 1 (CX3CL1) in native U251MG and U87MG cells after stimulation with control medium (m) or stem-like cell-conditioned medium (cm) supplemented with DMSO (control), TMZ [50 µM], AT101 [5 µM], or TMZ [50 µM] and AT101 [5 µM], respectively, for three and six days. CXCR7 expression in (**A**) U251MG cells and (**B**) U87MG cells; CXCL16 expression in (**C**) U251MG cells and (**D**) U87MG cells; CX3CL1 expression in (**E**) U251MG cells and (**F**) U87MG cells (*n* = 3). The significances between different stimulation approaches were determined with a two-way ANOVA test followed by Tukey’s multiple comparison test. Significant differences compared to the DMSO control are indicated directly above the bars and between different stimulations on a line linking the bars (* *p* < 0.05; ** *p* < 0.01; *** *p* < 0.001). The number represents the n-fold expression of the stimulation with stem-like cell-conditioned medium normalized to the stimulation with control medium. Error bars correspond to the standard deviation.

**Figure 8 ijms-22-03606-f008:**
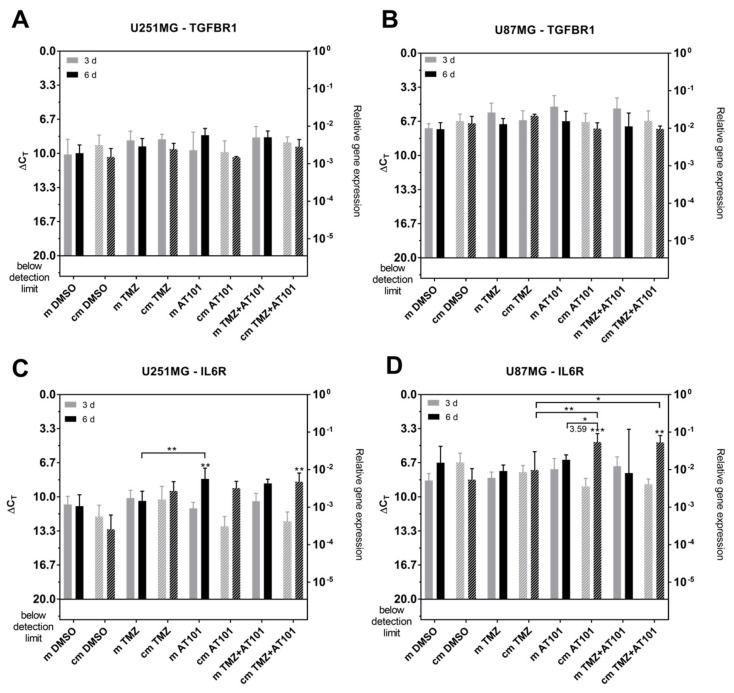
An interleukin 6 (IL-6) upregulation was detectable with stem-like cell conditioned media in U251MG cells only after combined treatment with TMZ plus AT101; in U87MG cells, both solely applied AT101 and combined TMZ plus AT101 treatment yielded an IL-6 upregulation under these treatment conditions. qPCR analysis of the mRNA expression of the cytokine receptors transforming growth factor beta receptor 1 (TGF-βR1) and IL-6 receptor (IL-6R) in native U251MG and U87MG cells after stimulation with control medium (m) or stem-like cell-conditioned medium (cm) supplemented with DMSO (control), TMZ [50 µM], AT101 [5 µM], or TMZ [50 µM] and AT101 [5 µM], respectively, for three and six days. TGF-βR1 expression in (**A**) U251MG cells and (**B**) U87MG cells; IL-6R expression in (**C**) U251MG cells and (**D**) U87MG cells (*n* = 3). The significances between different stimulation approaches were determined with a two-way ANOVA test followed by Tukey’s multiple comparison test. Significant differences compared to the DMSO control are indicated directly above the bars and between different stimulations on a line linking the bars (* *p* < 0.05; ** *p* < 0.01; *** *p* < 0.001). The number represents the n-fold expression of the stimulation with stem-like cell-conditioned medium normalized to the stimulation with control medium. Error bars correspond to the standard deviation.

## Data Availability

The data presented in this study are available within the article and in Appendix A.

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
