# Peer review of "Effects of the Anti-Tumorigenic Agent AT101 on Human Glioblastoma Cells in the Microenvironmental Glioma Stem Cell Niche"

_ijms, 2021, doi:10.3390/ijms22073606_

Round 1

Reviewer 1 Report

In this revised manuscript Caylioglu et al. described the effects of AT101 alone or in combination with temozolomide in glioblastoma cell lines and compared their effects in native (differentiated) and stem-like cells. Overall the manuscript is well written, the study was very well designed and the methods were adequately used. There are with only minor needs for editing. For example, the term “multiforme” is outdated in the nomenclature of glioblastoma – although this term was used only in the abstract, it should be avoided.

Concerning the results, while the demonstration of the efficacy of AT101 (alone and in combinations with temozolomide) in stem-like cells was observed, the results shown in Figure 3 are concerning. It is not clear how the authors explain the outstand increase in the % of cell death observed on the day 6 in the cells in the conditioned stem-cell media, observed in both cell lines. Could this increase in cell death be due to toxicity not related to the drug(s)? This needs to be clarified in the discussion.

Figure 7 is somewhat misleading. Although in the text the authors refer that AT101 increased the expression of IL6 in U87MG cell line only, the legend states simply that “AT101-treatment seemed to induce IL-6R expression”. Besides, on Figures 7C and 7D the differences marked with ### are comparisons of the same treatment at different time points, therefore are not very representative of the efficacy of the treatment compared to others. Importantly, this last note applies also to Figures 3A and 3B.

Finally, the authors did not state future directions for this important research. For example, are there plans to evaluate a possible synergy of AT101 with radiation? Are there plans for in vivo studies? These and other possibilities should be included in the discussion or conclusion section.

Reviewer 2 Report

Dear Members of the Editorial Board,

            Re: 1134856 - Review:  “Effects of the anti-tumorigenic agent AT101 on human    glioblastoma cells in the microenvironmental glioma stem cell niche”.

The authors show that U251MG and U87MG GBM cells grown as stem-like cells are sensitive to drug treatment of AT101 alone or a combination of AT101/Temozolomide.  The drug treatment promotes cell death and inhibits proliferation.  By contrast, naïve U251MG and U87MG GBM cells grown in regular media do not share the sensitivity to AT101 alone or a combination of AT101/Temozolomide.  However, when incubated in conditioned stem-like cell media, naïve U251MG and U87MG GBM cells become sensitive similar to their stem-like cell counterparts.  The authors found that ERKs lose their activity/phosphorylation upon the drug treatments paradigm and attribute that to cell death and loss of proliferation of U251MG and U87MG GBM cells in conditioned stem-like cell media.  Moreover, cytokines and their receptors are thought to play roles in drug-induced death and anti-proliferation.  Overall the manuscript requires major revisions for acceptance. 

Major Revisions:

  1. The authors have neglected to determine whether naïve U251MG and U87MG GBM cells in conditioned stem-like cell media undergo dedifferentiation and adopting a drug-sensitivity phenotype of U251MG and U87MG GBM cells in stem-like cells.The authors need to determine whether stem cell markers are up-regulated for naïve U251MG and U87MG GBM cells in conditioned stem-like cell media.

  1. The chemokine receptor CXCR7 and its chemokines appear not to play essential roles in adopting drug-sensitivity for naïve U251MG and U87MG GBM cells in conditioned stem-like cell media.However, IL6R and IL6 may have importance in anti-tumorigenesis.  If the IL6 signaling axis is relevant, the authors should add IL6 to naïve U251MG and U87MG GBM cells in regular media to determine if these Naïve cells become sensitive to AT101 alone or a combination of AT101/Temozolomide.

  1. Please define “inverse signaling.”

  1. For sentence “In native U87MG cells, which were treated with stem-like cell-conditioned media plus AT101 or TMZ plus AT101, both CXCL16 and CX3CL1 seemed to be downregulated (Figure 7D, F).”The reference is to Figure 6D, F.

  1. For sentences, “Thus, the microenvironment of stem-like cells, marked, besides others, by stem cell-secreted autocrine and active paracrine factors, seemed to trigger an enhanced AT101-chemosensitivity. Further, this effect was reflected by the inhibition of the ERK signaling pathway”. There are no Data presented in the manuscript to support stem cell-secreted autocrine and paracrine signaling factors. The sentences need to be revised or deleted.

Round 2

Reviewer 1 Report

I appreciate the authors’ responses to the comments and the way they modified the original manuscript to incorporate the suggestions. The figures in this version are cleaner and nicely explained.

My only suggestion at this point is that the authors could add the data concerning % dead cells in a table in the manuscript. The explanation in the letter of response was beautifully written. Summarizing all that information in a single table would make it readily available for the readers. Of course the same information is present at the figures, therefore is up to the authors and the editors considering this new modification.

Thank you for the opportunity to review this manuscript.

Reviewer 2 Report

The revised manuscript has addressed all of my concerns.